REGISTERED REPORT PROTOCOL

# Simultaneous separation of naproxen and 6-O-desmethylnaproxen metabolite in saliva samples by liquid chromatography–tandem mass spectrometry: Pharmacokinetic study of naproxen alone and associated with esomeprazole

**Thiago José Dionísio**[☯], **Gabriela Moraes Oliveira**[ID][☯], **Marina Morettin**[☯], **Flavio Cardoso Faria**[☯], **Carlos Ferreira Santos**[☯], **Adriana Maria Calvo**[ID]*[☯]

Biological Sciences, Bauru School of Dentistry, University of São Paulo, São Paulo, Brazil

☯ These authors contributed equally to this work.
* dricalvo@usp.br

## Abstract

Naproxen is a widely used non-steroidal anti-inflammatory drug for the control of postoperative inflammatory signs and symptoms in dentistry. Its association with esomeprazole has been widely studied and has yielded good results for the control of acute pain, even with the delayed absorption of naproxen owing to the presence of esomeprazole. To further understand the absorption, distribution, and metabolism of this drug alone and in combination with esomeprazole, we will analyze the pharmacokinetic parameters of naproxen and its major metabolite, 6-O-desmethylnaproxen, in saliva samples. A rapid, sensitive, and selective liquid chromatography-tandem mass spectrometric method for the simultaneous determination of naproxen and 6-O-desmethylnaproxen in saliva will be developed and validated. Sequential saliva samples from six patients will be analyzed before and 0.25, 0.5, 0.75, 1, 1.5, 2, 3, 4, 5, 6 8, 11, 24, 48, 72, and 96 h after the ingestion of one naproxen tablet (500 mg) and esomeprazole-associated naproxen tablets (500 + 20 mg), at two different times. After liquid-liquid extraction with ethyl acetate and HCl, the samples will be analyzed using an 8040 Triple Quadrupole Mass Spectrometer (Shimadzu, Kyoto, Japan). Separation of naproxen and its major metabolic products will be performed using a Shim-Pack XR-ODS 75Lx2.0 column and C18 pre-column (Shimadzu, Kyoto, Japan) at 40˚C using a mixture of methanol and 10 mM ammonium acetate (70:30, v/v) with an injection flow of 0.3 mL/min. The total analytical run time will be 5 min. The detection and quantification of naproxen and its metabolite will be validated, which elucidate the pharmacokinetics of this drug, thereby contributing to its proper prescription for the medical and dental interventions that cause acute pain.

**Data Availability Statement:** Data are available from the Mendeley database: (https://data.mendeley.com/datasets/zwsg6z5p8d/1), (DOI: 10.17632/zwsg6z5p8d.1)

**Funding:** We would like to thank São Paulo Research Foundation's (FAPESP) (Grant #2017/12725-0) for the financial support for this research in an appropriate field. The funders had and will not have a role in study design, data collection and analysis, decision to publish, or preparation of the manuscript.

**Competing interests:** The authors have declared that no competing interests exist.

# 1 Introduction

Naproxen is a non-steroidal anti-inflammatory drug of the propionic acid class widely used to manage chronic and acute pain, fever, swelling, and inflammation. Recently, a combination with esomeprazole has been introduced to the market to prevent gastrointestinal adverse reactions [1–3].

Naproxen ((S)-2-(6-methoxynaphthalen-2-yl)propanoic acid)is mainly metabolized by hydroxylation, forming 6-O-desmethylnaproxen, which then conjugates with glucuronic acid. Esomeprazole ((R)-5-Methoxy-2-(((4-methoxy-3,5-dimethylpyridin-2-yl)methyl)sulfinyl)-1H-benzo[d]imidazole) is the S-isomer of omeprazole, with gastric proton pump inhibitor activity. In the acidic compartment of parietal cells, esomeprazole is protonated and converted to the active achiral sulfonamide [4]. Several analytical methods have been developed to determine the concentrations of naproxen and 6-desmethylnaproxen using liquid chromatography–tandem mass spectrometry (LC-MS/MS) [5, 6], but none in saliva samples.

To study the pharmacokinetics (PK) of naproxen alone and in combination with esomeprazole in saliva samples and predict its functionality in cases of acute pain owing to delayed absorption, saliva samples will be collected several times after the administration of one pill of naproxen (500 mg) and naproxen combined with esomeprazole (500 mg + 20 mg), and concentrations of naproxen and its major metabolite 6-O-desmethylnaproxen will be determined. The use of saliva for PK tests makes collection simpler and more secure, without the need for a professional and with greater adherence to the study. Therefore, the aim of the present study was to disseminate the results of robust and reliable PK studies.

Additionally, the detection and quantification of naproxen and its metabolite will be validated to determine the PKs of this drug, thereby contributing to the development of a proper prescription for different medical and dental interventions that cause acute pain. The hypothesis of the present study is that the absorption of naproxen, when administered with esomeprazole, is slower than that when administered alone. Thus, the results of the study can be used to reassess the use of naproxen for pain control after medical and dental interventions that cause acute pain.

# 2 Material and methods

This project was approved by the Research Ethics Committee of the Bauru School of Dentistry/University of São Paulo (CAAE 49806115.0.0000.5417), registered in ClinicalTrials.gov (NCT03092193). All volunteers participating in this research will be fully informed about the project's content and procedures and sign a consent form.

## 2.1 Chemicals and reagents

Naproxen, 6-O-desmethylnaproxen, esomeprazole, and piroxicam (internal standard, IS) were purchased from Sigma-Aldrich (São Paulo, Brazil). Methanol, ammonium acetate, and other chromatographic grade chemicals used in the tests were purchased from Merck (Hohenbrunn, Germany). During all experiments, water from a Milli-Q Plus purification system (Millipore, Belford, MA, USA) was used.

## 2.2 Preparation of standard solutions

Stock solutions of naproxen (1 mg/mL methanol), 6-O-desmethylnaproxen (1 mg/mL methanol), esomeprazole (500 µg/mL methanol), and IS (1 mg/mL methanol) were prepared. A dilution series from each stock solution (10 µg/mL)was used to construct standard curves. When not in use, solutions were stored in the dark at -20°C and all stages of the research were

conducted under a sodium vapor lamp to avoid photodecomposition of naproxen, 6-O-des-methylnaproxen, esomeprazole, and the IS [7, 8].

## 2.3 LC-MS/MS

The concentrations of naproxen and its major metabolic, 6-O-desmethylnaproxen, were analyzed using an 8040 Triple Quadrupole Mass Spectrometer (Shimadzu, Kyoto, Japan) and piroxicam was used as an IS. Before performing the standard curve for naproxen, 6-O-desmethylnaproxen, esomeprazole, and the IS, the mobile phase was standardized to increase the reliability of the results that will be obtained from saliva samples in the PK studies [7]. As naproxen and esomeprazole ionize in different ways, only the PKs of naproxen and its major metabolite were investigated, as these are the drugs of most interest for pain control.

Initial analysis for drug characterization was performed by LC-MS/MS and separation was performed using a Shim-Pack XR-ODS 75Lx2.0 column and C18 pre-column (Shimadzu, Kyoto, Japan) at 40˚C using a mixture of methanol and 10 mM ammonium acetate (70:30, v/v) with an injection flow of 0.3 mL/min. The total analytical run time was 5 min.

The chromatographic effluent was directed to the 8040 Triple Quadrupole Mass Spectrometer (Shimadzu, Kyoto, Japan). Mass spectrometry was performed following the information obtained in the optimization of ions in positive/negative ionization mode using ionizing electrospray, with monitored selection for quantitative analysis. The voltage of the ionization electrospray capillary was 4.5 kV. Source and desolvation temperatures were maintained at 250˚C and 350˚C, respectively. Nitrogen was used as the mist gas at 3.0 L/min and argon gas as the collision gas at approximately 230 kPa. The cone voltage was defined for each transition and drug molecular ions were fragmented by the variable collision energy. All conditions were achieved by the direct injection of standard drug solutions of naproxen and 6-O-desmethylnaproxen without the separation column, resulting in transitions in which specific father/son ions were employed. The time of analysis for each event was 0.075 ms. The measurements were performed in multiple reaction monitoring mode to quantify the transitions presented in Fig 1. The peak areas for all components were integrated automatically using the LabSolutions software, version 5.97(Shimadzu, Kioto, Japan). The transitions considered for the quantification and qualification of the peaks are shown in Table 1.

## 2.4 Sample preparations

Drug-free saliva blanks (400 μL) were enriched with 5 ng of IS + 6.25 ng of naproxen and 6-O-desmethylnaproxen and 400 μL of 0.5 M HCl for the first calibration standard point. For the standard calibration curve, eight serial dilutions were performed and the final mass was equal to 0.024 ng of naproxen and 6-O-desmethylnaproxen. The IS mass was the same for all curve points. These standard calibration curve points were extracted with 2 mL of ethyl acetate in 15 mL polypropylene tubes, shaken vigorously for 15 min at high speed (Heidolph Multi Reax EU, Heidolph Instruments GmbH & Co., KG, Schawaback, Germany) and centrifuged at 2,500 g at 4˚C for 10 min. The organic phase (~1.8 mL) was then transferred and evaporated under airflow at room temperature in the chamber.

Saliva samples before naproxen alone and naproxen with esomeprazole intake (t = 0) were analyzed to ensure that there was no interference. After evaporation, the residues were dissolved in 100 μL methanol:10 mM ammonium acetate (70:30, v/v, isocratic), vortexed for 1 min, transferred to inserts, and then 5 μL of this solution was used for high-performance liquid chromatography.

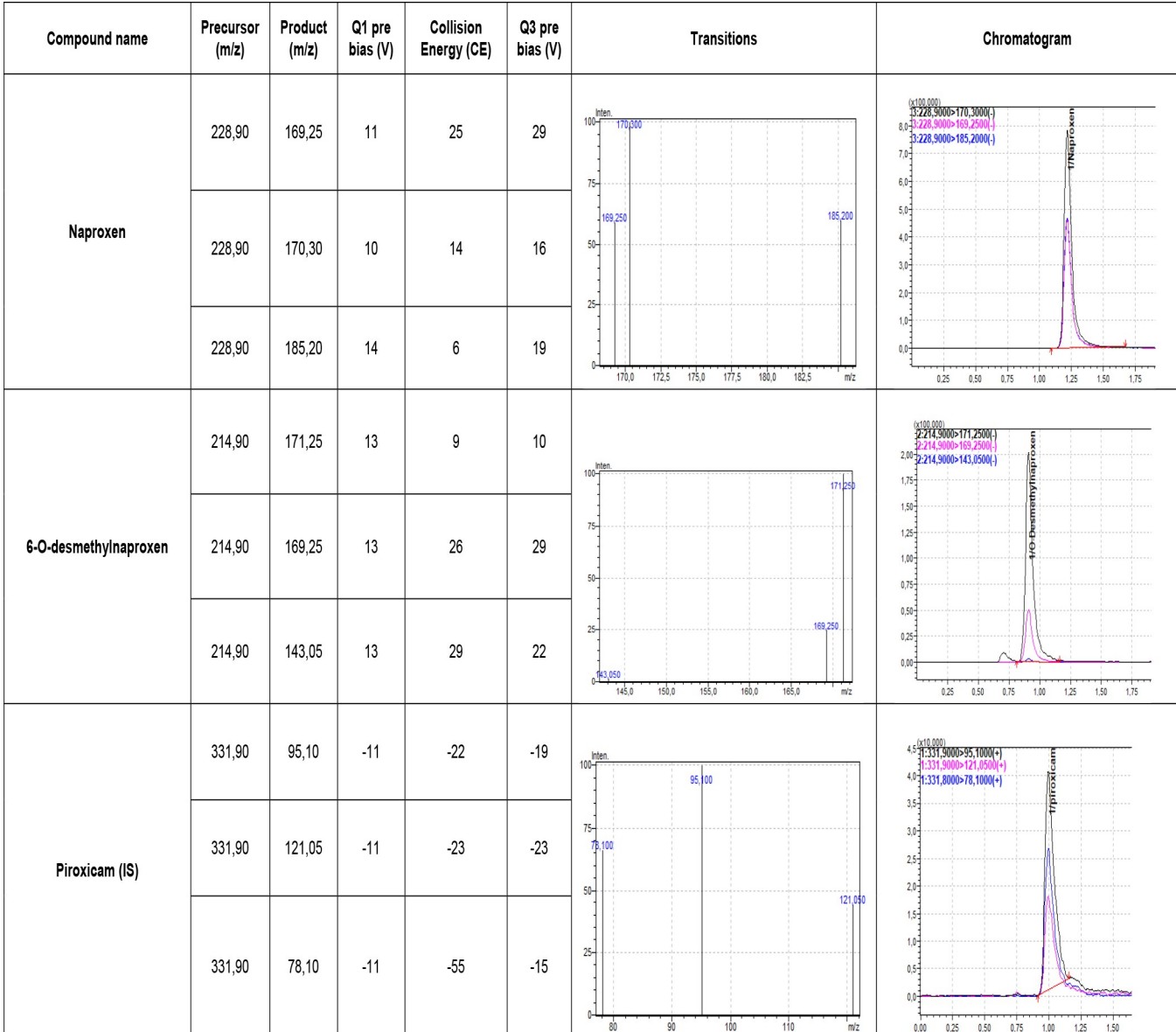

**Fig 1. Multiple reaction monitoring mode to quantify the naproxen, 6-O-desmethylnaproxen, and piroxicam transitions.**

## 2.5 Volunteer sample preparation

The PK study of naproxen and its major metabolite, 6-O-desmethylnaproxen, will be performed in six healthy volunteers, who will have sequential collections of saliva before and 0.25, 0.5, 0.75, 1, 1.5, 2, 3, 4, 5, 6 8, 11, 24, 48, 72, and 96 h after the ingestion of a naproxen tablet (500 mg) and esomeprazole-associated naproxen tablets (500 + 20 mg), at two different times, separated by 1–2 months. Samples will be centrifuged for 10 min (2,500 rpm) and the supernatant stored at -20°C until analysis.

The quantified PK parameters of naproxen and its major metabolite, 6-O-desmethylnaproxen, in saliva, alone and in combination with esomeprazole, will be estimated using the mean area under the curve, 0–72 h ($AUC_{0-72}$) and $AUC_{infinity}$, predicted total clearance (Clt/F), volume of distribution (Vd/F), and elimination half-life ($t_{1/2}$) based on concentrations

Table 1. Quantifier and qualifier product ions used in the analysis.

| Type | m/z | Inten | Act % | Idenf. Rang % |
|---|---|---|---|---|
| O-Desmethylnaproxen | | | | |
| T | 214.900>171.250[a] | 199246 | 100.00 | 100% |
| Ref 1 | 214.900>169.250[b] | 49605 | 24.90 | - |
| Ref 2 | 214.900>143.050 | 3283 | 1.65 | - |
| Naproxen | | | | |
| T | 228.900>170.30[a] | 513749 | 100.00 | 100% |
| Ref 1 | 228.900>169.250[b] | 336855 | 65.57 | - |
| Ref 2 | 228.900>185.200[b] | 288128 | 56.08 | - |
| Piroxicam | | | | |
| T | 331.900>95.100[a] | 33688 | 100.00 | 100% |
| Ref 1 | 331.900>121.050 | 15274 | 45.37 | - |
| Ref 2 | 331.900>78.100[b] | 23870 | 70.90 | - |

[a]Quantifier: the most intense peak.

[b]Qualifier: a second peak (SRM)t has been used as a qualifier for the same analysis.

obtained experimentally using a non-compartmental model with first-order elimination. These PK analyses will be performed using Phoenix WinNonlin® 8.1 software (Pharsight Corp., Mountain View, CA, USA). The data will be presented as mean ± standard deviation.

## 2.6 PK from six volunteers

In addition to validating the method for the detection and quantification of naproxen and its metabolite, 6-O-desmethylnaproxen, the PKs of six volunteers who will take one pill of naproxen (500 mg) and naproxen associated with esomeprazole (500 mg + 20 mg) will be investigated.

This study was registered as a clinical trial at ClinialTrials.gov (NCT030902193) after the analysis carried out by the website, as when prescribing these two drugs (naproxen alone and naproxen in combination with esomeprazole), a series of side effects may occur in a small portion of the population, such as allergies, gastric irritation, intestinal irritation, gastritis, colitis, and gastrointestinal bleeding. Such reports will be noted and analyzed during the clinical trial.

## 2.7 Analytical validation

The methodology for analyzing naproxen and 6-O-desmethylnaproxen in human saliva samples will be validated for essential parameters, such as residual effect and matrix effects, calibration curve, selectivity, precision, accuracy, stability, and dilution in accordance with the current guidelines of the United States Food and Drug Administration (October 2018) for bioanalytical methods to ensure reliability of the results obtained from the PK assays in human saliva.

Quality controls (QCs) for human saliva will be prepared by the addition of the final working solutions of treatment saliva to drug-free human saliva. Based on the lower limit of quantification (LLOQ), low-quality control (LQC), medium-quality control (MQC), and high-quality control (HQC) samples will be prepared. The QCs will be prepared, aliquoted (400 μL enriched saliva) in polypropylene tubes, and stored at −20˚C until analysis. For residual effect evaluation, injections of the blank sample before and after the injection of the HQC sample will be analyzed. The chromatogram of the white sample obtained after HQC injection will be compared with the sample chromatogram of the LLOQ. The residual effect will be considered

absent when the areas of interfering peaks eluted in the retention time of the analytes are less than 20% of the LLOQ and less than 5% of the IS area.

The matrix effect in the saliva will be evaluated using samples from six drug-free volunteers. Blank saliva samples from healthy volunteers will be obtained from the Laboratory of Clinical Pharmacology and Physiology, School of Dentistry, Bauru/USP. The matrix effect will be determined by comparing the peak areas obtained from naproxen, 6-O-desmethylnaproxen, and IS directly to the mobile phase (no matrix) with the peak in the presence of a matrix (white samples added to standard solutions of naproxen, 6-O-desmethylnaproxen, and IS after extraction). The normalizing factor of the IS in the matrix will be calculated for each sample matrix by dividing the ratio of the analyte by the response of the IS in the absence of the matrix. The coefficient of variation for the normalized IS must be less than 15%. Samples will be analyzed between the LQC and the HQC.

The saliva calibration curve will be prepared using the following concentrations in polypropylene tubes: 2500, 1250, 625, 312.5, 156.2, 78.1, 39.1, 19.5, 9.8, 4.9, and 2.4 ng/mL of naproxen and 6-O-desmethylnaproxen, plus a blank sample (no analyte and no IS) and a zero sample (IS only). Linear regression equations and correlation coefficients will be used from the ratios of the standard peak areas: IS plotted against the saliva samples used. Calibration curves will be plotted using the various concentrations of the IS versus the respective areas of their peaks, plotted as a linear regression line. Calibration curves will be approved when deviations of at least 75% of the calibration standards are less than or equal to 20% for LQC and 15% for other standards.

The LLOQ is defined as the lowest concentration of naproxen and 6-O-desmethylnaproxen quantified in saliva with acceptable precision and accuracy. The analyte interfering peaks at the retention time should be <20% of the analyte in the samples. To obtain the LLOQ, samples with the lowest detected concentration of naproxen, 6-O-desmethylnaproxen, and the IS in saliva will be evaluated ten times.

Three calibration curves for the analysis of naproxen and 6-O-desmethylnaproxen will be performed using aliquots of 400 μL of saliva containing a blank sample (no drug and no IS), one processed (IS added), and eight enriched with 25 μL standards of naproxen, 6-O-desmethylnaproxen, and IS, subjected to extraction and analysis procedures. The linearity will be evaluated using a linear mathematical model using the weighted equation $1/\chi 2$.

Accuracy and precision will be evaluated in a single assay (intra-assay) and three different assays (inter-assay, in three different periods). Five replicates of each assay were used for LLOQ, LQC, MQC, and HQC. Precision is expressed as a coefficient of variation (%) and accuracy by relative standard deviation (RSD), which must be between 15% and 20% (more tolerant) of the nominal value.

Naproxen and 6-O-desmethylnaproxen stabilities in saliva were measured using the LQC and HQC, analyzed in triplicate after preparation, and stored under different conditions (after short-term stability, post-processing stability, and three cycles of freezing and thawing). The concentrations of the samples will be interpolated with a recently prepared calibration curve. Samples will be considered stable when concentrations are within 15% of the nominal value. Additionally, samples in saliva will be added to create a concentration greater than the calibration curves for naproxen and 6-O-desmethylnaproxen. Then, this will be diluted five times to obtain a dilution integrity QC (DQC).

## 2.8 Statistical analysis

After analyzing the PK and pharmacodynamic results using the Phoenix WinNonlin software (version 8.1), the data will be organized descriptively and a paired *t*-test will be performed to

compare naproxen absorption after volunteers are administered, at two different times, naproxen alone or naproxen in combination with esomeprazole.

## Author Contributions

**Conceptualization:** Thiago José Dionísio, Carlos Ferreira Santos, Adriana Maria Calvo.

**Data curation:** Thiago José Dionísio, Gabriela Moraes Oliveira, Marina Morettin, Adriana Maria Calvo.

**Formal analysis:** Thiago José Dionísio, Gabriela Moraes Oliveira, Marina Morettin, Flavio Cardoso Faria, Adriana Maria Calvo.

**Funding acquisition:** Adriana Maria Calvo.

**Investigation:** Thiago José Dionísio, Flavio Cardoso Faria, Carlos Ferreira Santos, Adriana Maria Calvo.

**Methodology:** Thiago José Dionísio, Gabriela Moraes Oliveira, Marina Morettin, Flavio Cardoso Faria, Adriana Maria Calvo.

**Project administration:** Carlos Ferreira Santos, Adriana Maria Calvo.

**Resources:** Carlos Ferreira Santos, Adriana Maria Calvo.

**Software:** Thiago José Dionísio, Gabriela Moraes Oliveira, Marina Morettin, Adriana Maria Calvo.

**Supervision:** Flavio Cardoso Faria, Carlos Ferreira Santos, Adriana Maria Calvo.

**Validation:** Thiago José Dionísio, Carlos Ferreira Santos, Adriana Maria Calvo.

**Visualization:** Flavio Cardoso Faria, Carlos Ferreira Santos, Adriana Maria Calvo.

**Writing – original draft:** Thiago José Dionísio, Gabriela Moraes Oliveira, Marina Morettin, Flavio Cardoso Faria, Carlos Ferreira Santos, Adriana Maria Calvo.

**Writing – review & editing:** Thiago José Dionísio, Flavio Cardoso Faria, Carlos Ferreira Santos, Adriana Maria Calvo.

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
