## [Decision Letter · Decision Letter 0]

14 May 2020

PONE-D-20-08997

Simultaneous separation of naproxen and 6-O-desmethylnaproxen metabolite in saliva samples by LC MS / MS: PK study of naproxen associated or not with esomeprazole

PLOS ONE

Dear Dr Calvo,

Thank you for submitting your manuscript to PLOS ONE. After careful consideration, we feel that it has merit but does not fully meet PLOS ONE’s publication criteria as it currently stands. Therefore, we invite you to submit a revised version of the manuscript that addresses the points raised during the review process.

We would appreciate receiving your revised manuscript by Jun 28 2020 11:59PM. To enhance the reproducibility of your results, we recommend that if applicable you deposit your laboratory protocols in protocols.io, where a protocol can be assigned its own identifier (DOI) such that it can be cited independently in the future. For instructions see: http://journals.plos.org/plosone/s/submission-guidelines#loc-laboratory-protocols

We look forward to receiving your revised manuscript.

Kind regards,

Joseph Banoub, Ph,D., D. Sc.

Academic Editor

PLOS ONE

Reviewers' comments:

Reviewer's Responses to Questions

**Comments to the Author**

1. Does the manuscript provide a valid rationale for the proposed study, with clearly identified and justified research questions?

Reviewer #1: Partly

Reviewer #2: Partly

2. Is the protocol technically sound and planned in a manner that will lead to a meaningful outcome and allow testing the stated hypotheses?

Reviewer #1: Partly

Reviewer #2: Partly

3. Is the methodology feasible and described in sufficient detail to allow the work to be replicable?

Reviewer #1: No

Reviewer #2: No

4. Have the authors described where all data underlying the findings will be made available when the study is complete?

Reviewer #1: No

Reviewer #2: No

5. Is the manuscript presented in an intelligible fashion and written in standard English?

Reviewer #1: Yes

Reviewer #2: Yes

6. Review Comments to the Author

You may also provide optional suggestions and comments to authors that they might find helpful in planning their study.

Reviewer #1: This manuscript does not appear complete, as there are no analyses, results, or discussion sections. It appears to only be a partly-developed protocol as there is no mention of any planned statistical analyses for the data collected from 17 time points. This needs to be amended, and results should be shown if it is to be a true P/K study.

Additionally, it is mentioned that time point 0 for each drug were analyzed to ensure no interference, but there is no mention of how this analysis was performed.

Reviewer #2: The submission was not complete as only the abstract, introduction and methods were provided. Despite this, the following major issues were identified which must be addressed in the complete version:

• The method is intended for a clinical application among patients. Method validation is a must- it is not optional when dealing with patient samples. There was no mention of method validation. The authors must completely validate the method as per regulatory guidelines, the FDA, EMA or ICH.

• Though the authors indicate that this is the first time a method is reported for Saliva, did they attempt the conditions of the reported methods measuring the same compounds. Is there any similarities or improvements form an analytical stand point?

• Patient sample preparation [containing the drug] was not adequately described.

• PK studies are usually done in plasma- what information we can get from a Saliva for a medication intended for pain. Clarification and justification are needed to justify publication.

• What is the structures of the drugs and their monitored product ions? MSMS data should be shown along with the proposed structures. Why these specific product ions were chosen?

• Did the authors monitor the ratios of the quantifier/qualifier ions? A must when doing quantification with LC-MS/MS to ensure peak purity.

• In fact, it was not clear which product ion was the quantifier and which one is the qualifier

• The authors mention molecular ions. I think the authors mean protonated ions as molecular ions are the radical cations and it is not commonly seen in ESI.

• Why a structural analogue was used for internal standard instead of isotopically labelled IS.

7. PLOS authors have the option to publish the peer review history of their article (what does this mean?). If published, this will include your full peer review and any attached files.

Reviewer #1: No

Reviewer #2: No

---

## [Author Response · Author response to Decision Letter 0]

29 Jun 2020

Answer to reviewers

Reviewer #1: This manuscript does not appear complete, as there are no analyses, results, or discussion sections. It appears to only be a partly-developed protocol as there is no mention of any planned statistical analyses for the data collected from 17 time points. This needs to be amended, and results should be shown if it is to be a true P/K study.

Additionally, it is mentioned that time point 0 for each drug were analyzed to ensure no interference, but there is no mention of how this analysis was performed.

Answer: Thank you for your observation. The statistical analyses, which we intend to use, were added in the method section. Our article type is Registered Report Protocol, therefore, the results will be presented in the second manuscript as well as the discussion. The time point 0 refers to the collection of saliva from the volunteers before taking the medications. In fact, this collection is important to ensure that the pre-determined methods of detecting the drugs under study do not detect other molecules in the volunteers' saliva.

Reviewer #2: The submission was not complete as only the abstract, introduction and methods were provided. Despite this, the following major issues were identified which must be addressed in the complete version:

• The method is intended for a clinical application among patients. Method validation is a must- it is not optional when dealing with patient samples. There was no mention of method validation. The authors must completely validate the method as per regulatory guidelines, the FDA, EMA or ICH.

Answer: Thank you for your observation. Our article type is Registered Report Protocol, therefore, the results will be presented in the second manuscript as well as the discussion (if this manuscript is able to be accepted). We will we will execute the validations in all analyzes carried out in accordance with the FDA Please find the description of the experiments for validation in the methods section.

• Though the authors indicate that this is the first time a method is reported for Saliva, did they attempt the conditions of the reported methods measuring the same compounds. Is there any similarities or improvements form an analytical stand point?

Answer: Thank you for your question. Yes, we did a research where we compared the detection and quantification of Piroxicam in saliva and plasma. These results confirmed that in saliva it is possible to perform pharmacokinetic tests as in plasma (Calvo et al. J Pharm Biomed Anal. 2016 Feb 20;120:212-20. doi: 10.1016/j.jpba.2015.12.042).

• Patient sample preparation [containing the drug] was not adequately described.

Answer: Thank you for your observation. We adequately described patient sample preparation. Please find this new information in method section.

• PK studies are usually done in plasma- what information we can get from a Saliva for a medication intended for pain. Clarification and justification are needed to justify publication.

Answer: Thank you for your question. The idea of the study is to share the experience that it is possible to perform pharmacokinetic tests, using saliva, of any drug, including anti-inflammatory drugs. Pharmacokinetic studies are important for planning treatment according to the speed of metabolism of each person. This information was added in the introduction.

• What is the structures of the drugs and their monitored product ions? MSMS data should be shown along with the proposed structures. Why these specific product ions were chosen?

Answer: Thank you for your observation. Our mass spectrometer, based on the reported drug mass, automatically optimizes all parameters including product ions, which can be seen in Table 1. 

• Did the authors monitor the ratios of the quantifier/qualifier ions? A must when doing quantification with LC-MS/MS to ensure peak purity.

Answer: Thank you for your observation. These information is in the new table 2.

Table 2: Quantifier and qualifier product ion to be used in analysis.

O-Desmethylnaproxen

Type m/z Inten Act % Idenf. Rang %

T 214.900>171.250a 199246 100.00 100%

Ref 1 214.900>169.250b 49605 24.90 -

Ref 2 214.900>143.050 3283 1.65 -

Naproxen

Type m/z Inten Act % Idenf. Rang %

T 228.900>170.30a 513749 100.00 100%

Ref 1 228.900>169.250b 336855 65.57 -

Ref 2 228.900>185.200b 288128 56.08 -

Piroxicam

Type m/z Inten Act % Idenf. Rang %

T 331.900>95.100a 33688 100.00 100%

Ref 1 331.900>121.050 15274 45.37 -

Ref 2 331.900>78.100b 23870 70.90 -

aQuantifier - the most intense peak

bQualifier - a second peak (SRM)t have used as a qualifier for the same analysis

• In fact, it was not clear which product ion was the quantifier and which one is the qualifier

Answer: Thank you for your observation. Please check the answer above.

• The authors mention molecular ions. I think the authors mean protonated ions as molecular ions are the radical cations and it is not commonly seen in ESI.

Answer: Thank you for your observation. Sorry for the mistake. In fact we refer to the protonated ions.

• Why a structural analogue was used for internal standard instead of isotopically labelled IS.

Answer: Thank you for your question. Previous studies by our group have shown that Piroxicam could be used successfully as an internal standard. However, we have the conscience and the intention to start using the isotopically labeled IS. During the validation process, we will guarantee that there will be no interference from the chosen IS.

Answer to editors

1. We suggest you thoroughly copyedit your manuscript for language usage, spelling, and grammar. If you do not know anyone who can help you do this, you may wish to consider employing a professional scientific editing service. 

● The name of the colleague or the details of the professional service that edited your manuscript.

Answer: Thank you for your suggestion. We contracted the correction and editing service from Editage (https://www.editage.com.br). Follows the certificate issued by the company attached in cover letter. 

Answer: Thank you for your suggestion. A track changed manuscript was uploaded as supporting information.

Answer: Thank you for your suggestion. A clean copy of the edited manuscript was uploaded as manuscript file.

We are grateful for São Paulo Research Foundation’s (FAPESP) financial support. Grant #2017/12725-0.

"The funders had and will not have a role in study design, data collection and analysis, decision to publish, or preparation of the manuscript."

Answer: Thank you for your observation. We removed funding information from the acknowledgements. The currently Funding Statement is correct. We would like to thank FAPESP (Grant #2017/12725-0) for the financial support for this research in an appropriate field. 

3. To comply with PLOS ONE submission guidelines, in your Methods section, please provide additional information regarding your statistical analyses that will be conducted in your study. For more information on PLOS ONE's expectations for statistical reporting, please see https://journals.plos.org/plosone/s/submission-guidelines.#loc-statistical-reporting

Answer: Thank you for your suggestion. Please find the following statement in method section. After analyzing the PK and pharmacodynamic results using the Phoenix WinNonlin software (version 8.1), the data will be organized descriptively and a paired t-test will be performed to compare naproxen absorption after volunteers are administered, at two different times, naproxen alone or naproxen in combination with esomeprazole.

Answer: Thank you for your observation. We do not wish to make changes to our data availability statement.

5. It appears that your ORCiD iD has not been validated in your Editorial Manager account and we are unable to proceed until that step is complete.

Answer: Thank you for your observation. We validated the ORCiD iD in Editorial Manager.

---

## [Editor Report · Decision Letter 1]

6 Jul 2020

Simultaneous separation of naproxen and 6-O-desmethylnaproxen metabolite in saliva samples by LC MS / MS: PK study of naproxen associated or not with esomeprazole

PONE-D-20-08997R1

Dear Dr. Calvo,

We’re pleased to inform you that your manuscript has been judged scientifically suitable for publication and will be formally accepted for publication once it meets all outstanding technical requirements.

Kind regards,

Joseph Banoub, Ph,D., D. Sc.

Academic Editor

PLOS ONE
---

## [Editor Report · Acceptance letter]

24 Jul 2020

PONE-D-20-08997R1 

Simultaneous separation of naproxen and 6-O-desmethylnaproxen metabolite in saliva samples by liquid chromatography–tandem mass spectrometry: pharmacokinetic study of naproxen alone and associated with esomeprazole 

Dear Dr. Calvo:

I'm pleased to inform you that your manuscript has been deemed suitable for publication in PLOS ONE. Congratulations! Your manuscript is now with our production department. 

Kind regards, 

on behalf of

Dr. Joseph Banoub 

Academic Editor

PLOS ONE